

# Mechanical loading regulates osteogenic differentiation and bone formation by modulating non-coding RNAs

Huili Deng[1] and Dongfeng Wan[2]

[1] School of Medicine, Xiamen University, Xiamen, Fujian Province, China
[2] School of Health, Shanghai Normal University Tianhua College, Shanghai, China

## ABSTRACT

Bone tissue is highly responsive to mechanical stimuli, with mechanical loading serving as a crucial regulator of bone formation and resorption. The cellular transduction of mechanical loading involves intricate mechanisms, prominently featuring non-coding RNAs (ncRNAs). Various ncRNAs, including long non-coding RNAs (lncRNAs) and microRNAs (miRNAs), collaboratively regulate pathways involved in bone formation under mechanical loading. This article elucidates the mechanisms by which mechanical loading influences bone formation through ncRNAs, summarizing key ncRNAs and their regulatory pathways. Aimed at researchers and clinicians in molecular biology, orthopedics, and regenerative medicine, this study provides a theoretical foundation for the future application of mechanical loading to regulate osteogenic differentiation and offers insights into treating diseases associated with abnormal bone formation.

## INTRODUCTION

Components of bone tissue are hard and dense connective tissue (*de Buffrénil & Quilhac, 2021*). Bone is always in a dynamic mechanical environment throughout life. Bone is constantly reshaped by bone formation and bone resorption, and this reshaping enables the bone to accommodate changing loads and maintain homeostasis (*Salhotra et al., 2020*). As a typical mechanically responsive tissue, bone can adapt to mechanical stimuli, where appropriate physiological loading induces bone formation, while under- or over-loading may lead to bone resorption (*Roberts & Huja, 2016*). Mechanical loading plays a vital role in maintaining the balance between bone formation and bone resorption (*Wang et al., 2022a*). Bone reconstruction has been reported to be modulated by diverse mechanical stresses such as fluid shear stress (FSS), strain, distraction stress, compressive stress, and microgravity (MG) (*Thompson, Rubin & Rubin, 2012*). These mechanical stimuli are sensed by mechanosensitive cells associated with bone reconstruction, including osteoblasts, osteoblasts (OBs), osteoclasts (OCs), chondrocytes, and mesenchymal stem cells (MSCs) (*Dong et al., 2021*). These cells are intertwined in space and time, creating a synergy that responds to mechanical stimuli and translates them into biochemical signals that regulate the processes of bone formation and resorption.

Corresponding author
Dongfeng Wan, derfulwon@126.com

Intracellular transduction of mechanical stimuli is a comprehensive process: mechanical stimuli are first sensed by different cellular structures, such as integrins and the cytoskeleton (*Frith et al., 2018*), and subsequently delivered to apical structures, including the actin cortical reticulum and the plasma membrane, and then to intracellular regions, ultimately activating various types of mechanosensitive signaling pathways, such as the $Ca^{2+}$ signaling pathway, the BMP2 signaling pathway, and so on. These signaling pathways subsequently convert mechanical stimuli into biochemical signals that regulate the processes of osteoblast growth, migration, and differentiation (*He et al., 2021*). In addition to the coordinated control of multiple factors such as hormones, signaling factors, and environmental factors, non-coding RNAs (ncRNAs) play a vital role in the transduction of mechanical signals. NcRNA is a unique RNA transcript that does not encode proteins (*Kaikkonen & Adelman, 2018*). Studies have shown that most of the more than 80% of the transcribed human genome is transcribed as ncRNA (*Anastasiadou, Jacob & Slack, 2018*). NcRNAs mainly include small nucleolar RNA (snoRNA), circular RNA (circRNA), microRNA (miRNA), and lncRNA, of which miRNA and lncRNA are the most studied types (*Ali et al., 2021*).

During bone metabolism, lncRNAs may influence cellular activity by regulating gene transcription through interactions with miRNAs in OCs and OBs (*Liu et al., 2017b*). Several studies have shown that ncRNAs, especially mechanoresponsive ncRNAs, play critical roles in a variety of biological and physiological contexts such as mechanotransduction and promotion of osteogenic differentiation (*Hu et al., 2017*; *Qin et al., 2021*). Differentiation of OBs is crucial in the process of bone formation. Normally, most of the bone matrix is mineralized by osteoblasts, but only mature OBs can complete this process. In addition, secretion and calcification of osteoid bone likewise require the participation of mature OBs. Studies have shown that osteoblast differentiation is triggered by the secretion of active substances such as hormones and growth factors induced by mechanical stimulation. Mechanical stimulation plays an important role in the regulation of osteogenic differentiation (*Wang et al., 2017*). However, the specific mechanisms by which mechanical stimulation regulates osteogenic differentiation remain unclear, especially the key ncRNA that responds to mechanical stimulation during osteogenic differentiation and their specific regulatory pathways. Therefore, this article provides a comprehensive summary of the key ncRNAs and their pathways involved in the regulation of mechanical load during osteogenic differentiation by reviewing relevant research articles on different types of mechanical loading affecting cellular osteogenic differentiation in recent years. The main purpose of this study is to clarify the specific intrinsic mechanism by which mechanical loading regulates bone formation by modulating nRNAs. Therefore, we hope to provide a theoretical basis for the further application of mechanical loading to promote osteogenic differentiation in the future, as well as to provide relevant ideas for the treatment of related bone formation abnormalities and other related diseases.

## SURVEY METHODOLOGY

We used the PubMed Advanced Search Builder to systematically search for articles with the following combinations of titles/abstracts: ("mechanical loading" OR "microgravity"

OR "fluid shear stress" OR "mechanical stretch" OR "mechanical stress" OR "mechanotransduction") AND ("bone formation" OR "osteogenesis" OR "bone remodeling") AND ("non-coding RNA" OR "ncRNA" OR "microRNA" OR "miRNA" OR "long non-coding RNA" OR "lncRNA" OR "circular RNA" OR "circRNA"). Articles not related to the role of mechanical loading in bone formation through non-coding RNAs were excluded. Our search was not refined by publication date, journal, or impact factor of the journal, authors, or authors' affiliations. Additionally, we reviewed relevant guidelines and reports from key organizations in the field of bone biology and mechanotransduction.

## OVERVIEW OF ncRNAs

Open reading frames (ORFs) are genome segments that may encode proteins. Canonical ORFs are typically well-translated, producing proteins with specific biological roles (*Couso & Patraquim, 2017*). Traditionally, ncRNAs are considered non-coding due to the need for start codons and the size limitations of small ORFs (sORFs) (*Jackson et al., 2018*). However, recent studies demonstrate that ncRNAs are not mere transcriptional noise. They act as key connectors in genetic networks, influencing protein effectors and playing crucial roles in cellular responses and cell fate determination. NcRNAs regulate gene expression in development, physiological functions, and disease progression (*Frith et al., 2018*; *Wu et al., 2019*). Research indicates that pri-miRNA and lncRNA have 5′ terminal modifications and 3′ polyadenylation tails, suggesting potential protein-coding capabilities (*Fazi & Fatica, 2019*). Although circRNA lacks these terminal features, it can initiate translation through internal ribosome entry sites (IRESs) or m6A modifications (*Legnini et al., 2017*; *Yang et al., 2017*). Studies have shown that changes in ncRNA expression, influenced by external factors, can regulate bone metabolism in osteoporosis by affecting osteoblast and osteoclast activity, thus impacting bone formation (*Guo et al., 2022*; *Zhou et al., 2021a*).

### MiRNA

MiRNA is a short ncRNA, about 22 nucleotides long, that regulates protein expression by interacting with target mRNA to control its degradation and translation (*Zhu et al., 2020*). MiRNAs function through peptide or protein-coding and RNA-based mechanisms (*Bakhti & Latifi-Navid, 2022*). For example, pri-miRNAs of miR-200a and miR-200b encode peptides affecting tumor development, while their mature forms regulate cancer progression by targeting the 3′ UTRs of genes. MiRNA regulation involves various mechanisms, including regulatory RNAs that act as "sponges". These sponges, or competing endogenous RNAs (ceRNAs), bind miRNAs to buffer their activity (*Li et al., 2021*). Such sponges include pseudogenes, lncRNAs, and circRNAs, which interact with miRNAs to modulate protein production (*Alkan & Akgül, 2022*). For instance, lncRNA TUG1 acts as a "sponge" for miR-222-3p, inhibiting its negative regulation of Smad2/7, thus promoting osteogenic differentiation of periodontal ligament stem cells (PDLSCs) (*Wu et al., 2020*). In osteogenic differentiation, miRNAs target genes like Runx2, BMP, Smad, TGF-β, and BMPR pathways (*Zhang et al., 2018*).

## CircRNA

CircRNA is an ncRNA molecule with a closed-loop structure that is resistant to degradation by RNA enzymes (*Zhao et al., 2022*). CircRNAs, like miRNAs, can function by encoding peptides/proteins or RNA-based mechanisms. It was demonstrated that circRNA are engaged in bone remodeling regulation by acting as molecular sponges through the miRNA-mRNA axis (*Chen et al., 2019*; *Qian et al., 2017*). *Chen et al. (2019)* demonstrated that the circRNA_28313/miR-195a/CSF1 axis is crucial for osteoclast differentiation and bone resorption in mice. CircRNA_28313 is upregulated in bone marrow monocyte/macrophage (BMM) cells in response to RANKL and CSF1 treatment, promoting osteoclast differentiation. Mechanistically, circRNA_28313 functions as a competing endogenous RNA (ceRNA), sequestering miR-195a and alleviating its suppression of CSF1, thus facilitating osteoclast differentiation, as confirmed by luciferase reporter and RIP assays. Furthermore, during BMP2-induced osteogenic differentiation of MC3T3-E1 cells, RNA-seq analysis identified 158 differentially expressed circRNAs, such as circRNA.5846, circRNA.19142, and circRNA.10042, which were significantly upregulated (*Qian et al., 2017*). These circRNAs act as miRNA sponges, interacting with miRNAs like miR-7067-5p, thereby regulating the circRNA-miRNA-mRNA network. This interaction influences key signaling pathways, including FGF, EGF, PDGF, and Wnt, essential for osteoblast differentiation (*Majidinia, Sadeghpour & Yousefi, 2018*; *Vlashi et al., 2023*). In summary, different functions of target mRNAs in the miRNA-mRNA axis determine the role of miRNAs and circRNAs in modulating osteoblast differentiation.

## LncRNA

LncRNAs are non-coding transcripts over, 200 nucleotides long, primarily transcribed by RNA Polymerase II and located in the nucleus or cytoplasm (*Mattick et al., 2023*). Their secondary structures provide binding sites for proteins and RNAs, regulating transcription, translation, cell differentiation, and other processes (*Li, Zhu & Luo, 2016*). LncRNAs influence gene expression at transcriptional, post-transcriptional, and epigenetic levels, impacting cell growth, cycle control, and differentiation (*Fu et al., 2017*; *Huang et al., 2018*; *Terracciano et al., 2017*). Research indicates that lncRNAs play roles in osteocyte physiology and are linked to skeletal disorders like osteoporosis (*Lee et al., 2021*; *Li et al., 2018*; *Man et al., 2021*). For instance, inhibiting lncRNA AK016739 *in vivo* enhances osteogenic gene expression and bone formation in ovariectomized mice (*Yin et al., 2019*). The subcellular localization of lncRNAs affects their regulatory mechanisms. Intranuclear lncRNAs are involved in transcriptional activation and epigenetic control, while cytoplasmic lncRNAs regulate post-transcriptional processes by acting as ceRNAs (*Thomson & Dinger, 2016*). Unlike miRNAs and circRNAs, lncRNAs can encode peptides or function as RNA molecules, but not both. For example, the peptide from lncRNA HOXB-AS3 inhibits tumor development, while the lncRNA itself lacks tumor suppressor functions (*Wang et al., 2019*).

In summary, ncRNAs are crucial regulators of gene expression in development, physiology, and disease, particularly in cellular osteogenic differentiation. MiRNAs and circRNAs can regulate bone formation through both peptide/protein-coding and

RNA-based mechanisms, while lncRNAs function solely as RNA molecules. The role of lncRNAs in osteoblasts depends on their subcellular localization: intranuclear lncRNAs are involved in transcriptional activation and epigenetic control, whereas cytoplasmic lncRNAs regulate post-transcriptional processes by acting as ceRNAs. They bind to miRNAs, upregulating miRNA target genes, and modulating osteogenic differentiation.

## LncRNA-miRNA-mRNA REGULATORY NETWORK UNDER MECHANICAL LOADING CONDITIONS

Mechanical loading is a crucial epigenetic factor in bone tissue regeneration (*Ma et al., 2023*). Organisms use mechanoreceptors like adhesive plaques, the cytoskeleton, and membrane channels to sense extracellular mechanical stress and convert it into intracellular biochemical signals. Mechanical stress regulates osteogenic differentiation by modulating ncRNA expression. For instance, it targets genes related to bone formation and resorption, such as ACVR2B, Runx2, and Hmga2, through miRNAs like miR-21, miR-103a, and miR-214 (*Chen et al., 2017*; *Yuan et al., 2017*). This regulation influences bone remodeling *via* BMP and Wnt/PCP signaling pathways. *Cai et al. (2022)* constructed force-related lncRNA and ceRNA networks, identifying lncRNA-transcription factor loops that highlight lncRNAs' regulatory roles. They analyzed differentially expressed mRNAs (DE mRNAs) from the GEO dataset (GSE112122) using edgeR. For the ceRNA network, ENCORI predicted lncRNA targets of force-sensitive miRNAs, identifying miRNA-targeted mRNAs by intersecting miRTarBase, miRDB, and TargetScan data. Cytoscape visualized lncRNA–miRNA–mRNA pairs. FR lncRNA expression was validated *via* qRT-PCR on mechanically stretched C3H/10T1/2 cells. Additionally, Anthrax Toxin Receptor 1 (Antxr1) is a mechanosensitive protein that promotes chondrogenic differentiation in BMSCs (*Cheng et al., 2019*; *Feng et al., 2023*). *Liu et al. (2024)* found that mechanical forces mediate the Antxr1/lncRNA H19/Wnt/β-catenin axis, affecting osteogenic differentiation and regulating lncRNA H19 expression.

Mechanical-sensitive lncRNAs and miRNAs regulate gene expression by forming lncRNA-miRNA-mRNA networks in response to mechanical stress, influencing osteogenic differentiation. Recent studies have increasingly focused on these networks, highlighting the relationships between mechanical loading, ncRNAs, and osteogenic differentiation. *Wang et al. (2022b)* that dynamic tension in PDLSCs led to differential expression of 344 lncRNAs, 57 miRNAs, 41 circRNAs, and 70 mRNAs, primarily enriched in processes related to osteogenesis and mechanical stress. Functional enrichment using DAVID identified GO terms and KEGG pathways linked to cell differentiation and calcium ion transport. A PPI network constructed with STRING and visualized in Cytoscape identified key modules and hub genes, including EGR1, RIPK4, and ATF3, which are critical for osteogenic differentiation and stress response (*Wang et al., 2022b*). In another study, *Wang et al. (2021a)* constructed a network based on differentially expressed lncRNAs and miRNAs in stretched PDLSCs. It showed that single lncRNAs can regulate multiple miRNAs, affecting mRNA expression and osteogenic differentiation. Notably, hsa-miR-21 and hsa-miR-4492 were upregulated, interacting with lncRNA TCONS_0018927, with hsa-miR-21 targeting ACVR2B to facilitate differentiation. In
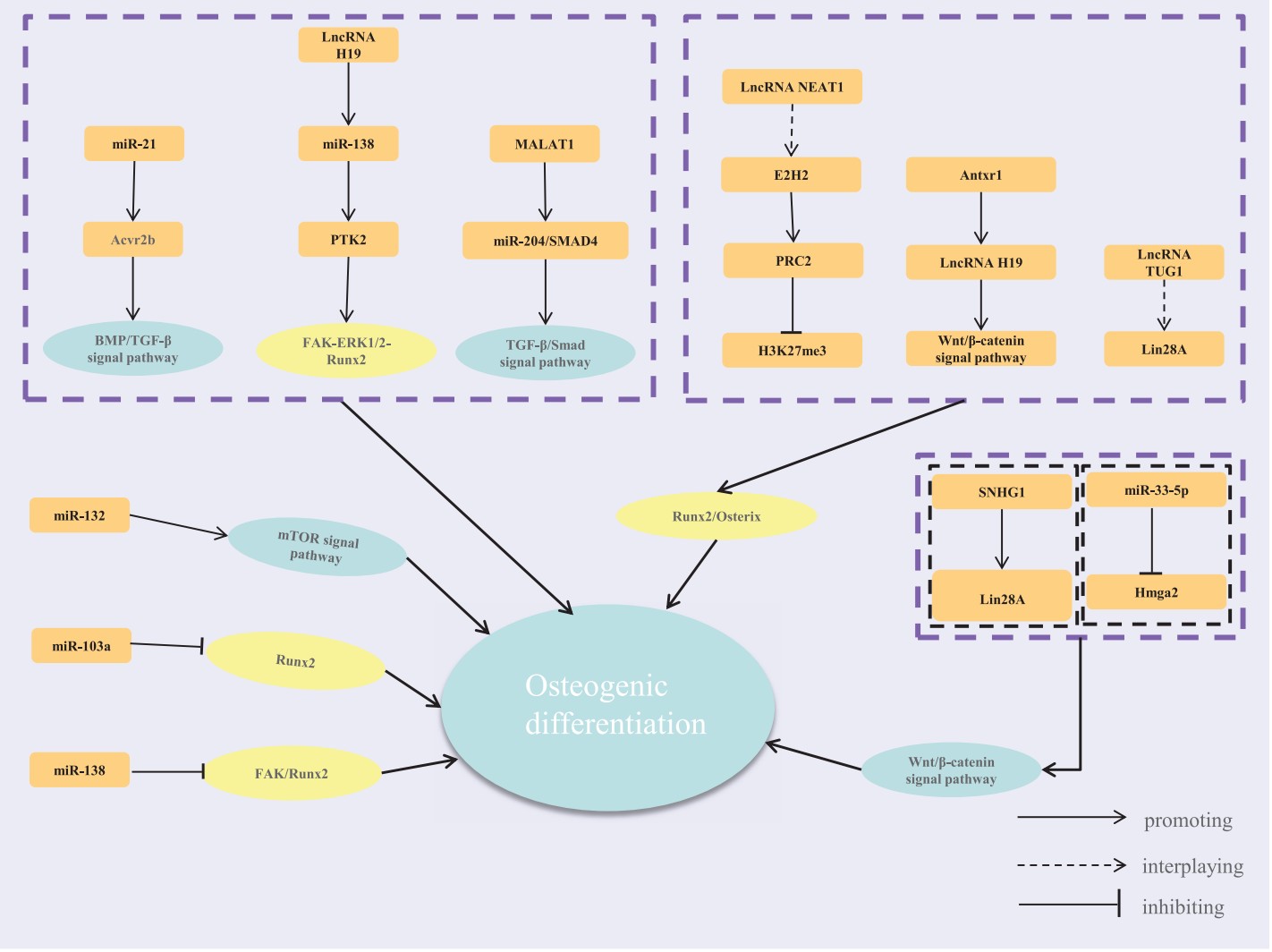

**Figure 1 The regulatory network of lncRNA-miRNA-mRNA.**

addition to conventional lncRNA-miRNA-mRNA or ceRNA networks, transcription factors (TFs) can also be regulated by ceRNA interactions at the post-transcriptional level. Once in the nucleus, TFs regulate gene transcription and establish feedback loops with lncRNAs (*Ji et al., 2020*).

In summary, cells perceive extracellular mechanical stimuli through adhesive plaques and the cytoskeleton under mechanical loading. These signals are transmitted *via* signaling pathways involving transcription factors and ncRNAs. MiRNAs and lncRNAs are key players in responding to mechanical signals, forming complex regulatory networks crucial for osteoblast proliferation, differentiation, and bone tissue remodeling (Fig. 1). While current studies highlight these networks' roles in bone metabolism, different mechanical loadings may activate distinct pathways, leading to varied network responses. Future research should compare these changes to uncover the commonalities and specificities of mechanical regulatory networks. This understanding could

provide new strategies and targets for treating diseases like osteoporosis and periodontal disease.

## MECHANISMS OF ncRNA REGULATION OF OSTEOGENIC DIFFERENTIATION UNDER DIFFERENT MECHANICAL LOADING

Bone tissue adapts to mechanical stress through a feedback mechanism (*Wang et al., 2022a*). Changes in local mechanical stress send signals to bone cells, affecting the activity, proliferation, or migration of osteogenesis-related cells. This alters the balance between bone formation and resorption, leading to structural changes to meet new demands (*Yavropoulou & Yovos, 2016*). Appropriate mechanical stimuli promote osteogenic differentiation, while extreme conditions like overloading or unloading can inhibit it, causing bone resorption and potentially leading to osteoporosis and other metabolic diseases (*Goetzke et al., 2018*). For instance, reduced gravity decreases osteogenic differentiation and increases adipogenic differentiation in human MSCs. The type, duration, and intensity of mechanical stimuli influence cellular signaling pathways and mechanosensitive ncRNA regulatory networks, leading to significant variations (*Fang, Wu & Birukov, 2019*). Various mechanical loadings, such as microgravity, fluid shear stress, tensile stress, compressive stress, and vibration, modulate osteogenic differentiation (*Liu et al., 2022b*). This article summarizes the mechanisms by which these mechanical loadings regulate osteogenic differentiation, offering insights for treating osteogenesis-related diseases.

### Mechanism of ncRNA regulation of osteogenic differentiation under MG condition

MG is a special and relatively thorough mechanical unloading environment that can quickly lead to massive bone loss (*Prasad et al., 2020*). It was shown that bone loss resulting from mechanical unloading is characterized by a decoupling of bone turnover: decreased bone formation while increased bone resorption (*Juhl et al., 2021*). This process primarily occurs in patients requiring prolonged immobilization and in astronauts under MG conditions. During spaceflight, may experience a bone density decrease of up to 2% of their total bone mass per month, which is comparable to the bone loss of postmenopausal women in more than 1 year (*Lescale et al., 2015*). Many investigators have suggested that the suppression of bone formation, as a result of mechanical unloading, is caused by a reduction in osteoblast activity. It is necessary and feasible to develop promising bone formation strategies by studying the molecular regulatory mechanisms of osteoblast function under unloading environments.

Genomics data analysis showed that osteoblasts changed significantly in the expression of hundreds of genes after MG exposure. Among them, lncRNAs play a crucial role in osteoblast differentiation under simulated MG environments (*Hou et al., 2018*). *Hu et al. (2017)* found that the expression of 857 lncRNAs changed significantly in MC3T3-E1 cells following the simulation of MG exposure. *Liu et al. (2022a)* studied mechanosensitive lncRNAs in osteoblasts under simulated MG conditions and found a significant reduction

in the levels of lncRNA nuclear enrichment transcript 1 (Neat1). Subsequent *in vivo* experiments demonstrated that Neat1 knockout mice had an impaired skeletal phenotype and a diminished response to mechanical loading and hindlimb unloading, resulting in impaired bone formation, decreased bone structure and strength, and reduced bone mass (*Liu et al., 2022a*). LncRNA Neat1 is the core structural component of paraspeckles, which are subnuclear bodies formed in the interchromatin space (*Todorovski, Fox & Choi, 2020*). These membrane-free structures are involved in the regulation of gene expression by sequestering specific RNAs and proteins. Paraspeckles facilitate the nuclear retention of certain mRNAs, such as Smurf1, thereby preventing their translation in the cytoplasm (*Yamazaki et al., 2021*). This retention inhibits the expression of Smurf1 target genes, which are regulated through ubiquitination. Smurf1 is a crucial E3 ubiquitin ligase necessary for maintaining skeletal homeostasis, and its overexpression hinders skeletal development (*Sun et al., 2018*). Consequently, the degradation of Runx2 is inhibited, enhancing osteoblast function. Mechanical loading induces the upregulation of LncRNA Neat1, promoting the assembly, solidification, and extension of paraspeckles, thereby enhancing osteoblast function. Therefore, mechanical loading induces the upregulation of LncRNA Neat1 expression as well as the assembly, solidification, and extension of paraspeckles to enhance osteoblast function. Conversely, mechanical unloading inhibits Neat1 expression, reduces osteoblast function, and thus inhibits osteogenic differentiation. In another study, mechanical unloading promoted the expression of lncRNA ODSM, and then partially reduced apoptosis of MC3T3-E1 and promoted their differentiation. Unlike lncRNA Neat1, lncRNA ODSM is localized in the cytoplasm and is mainly engaged in post-transcriptional regulation of gene activity. *Wang et al. (2020)* demonstrated that lncRNA ODSM can either directly regulate the miR-139-3p target gene ELK1 (a microgravity-sensitive protein) or function as a "sponge" for miR-139-3p to indirectly regulate ELK1, leading to influence osteogenic differentiation (*Wang et al., 2018*). Several studies reported that miRNAs could regulate the osteogenic differentiation process by targeting Runx2 within osteoblasts. *Zhou et al. (2021b)* showed that miR-133a could target Runx2 to positively regulate osteoblast activation and mineralization, and significantly improve bone loss, microarchitecture, and biomechanical properties in a mouse hindlimb-unloaded (HU) model. In another study, miR-103a can negatively regulate Runx2. Therapeutic inhibiting miR-103a could partly restore bone loss induced by mechanical unloading (*Chen et al., 2020*). In addition, the BMPR2 pathway is also involved in osteogenic differentiation under conditions of mechanical unloading. It was shown that C2C12 cells, a multifunctional mesenchymal progenitor cell, had significantly increased expression of miR-494 when exposed to MG conditions. MiR-494 modulates BMP signaling by directly targeting BMPR2 and RUNX2, thereby inhibiting BMP-induced osteogenic differentiation at multiple levels (*Qin et al., 2019*). In MC3T3-E1 cells and primary osteoblasts, mechanical unloading can lead to reduced miRNA-33-5p and miRNA-132-3p expression (*Zhang et al., 2023*), and thus affect their osteogenic differentiation. Further studies have been conducted to explore the expression profiles and functional networks of circRNAs in MC3T3-E1 under an MG environment. *Cao et al. (2021)* found that circ-014154 was engaged in the osteogenic differentiation of MC3T3-E1

**Table 1 Mechanism of ncRNA regulation of osteogenic differentiation under MG conditions.**

| References | Sample resources | Targets | NcRNAs |
|---|---|---|---|
| Liu et al. (2022a) | Osteoblasts | Runx2 | Neat1 |
| Wang et al. (2020) | MC3T3-E1 | ELK1 | lncRNA ODSM/miR-139-3p |
| Wang et al. (2018) | MC3T3-E1 | ELK1 | miR-139-3p |
| Wang et al. (2016) | MC3T3-E1 | Runx2 and Hmga2 | miRNA-33-5p |
| Hu et al. (2015) | MC3T3-E1 | Runx2 and Ep300 | miRNA-132-3p |
| Qin et al. (2019) | C2C12 cell and HEK 293T cell | BMPR2 and RUNX2 | miR-494 |
| Zhou et al. (2021b) | MC3T3-E1and BMMs | Runx2 and Osx | miR-133a |
| Cao, Lv & Lv (2015) | hMSCs | BMPR2 | miR-153 |
| Bergh et al. (2006), Cao et al. (2021), Sun et al. (2015a, 2015b) | MC3T3-E1 cell | LTCCs and Cav1.2 LTCC | miR-103 |

under a simulated MG condition, and its expression increased with time over 72 h. In addition, it was shown that lots of circRNAs were partially involved in osteogenic differentiation regulation as ceRNAs.

In conclusion, the mechanical unloading environment provided by MG can lead to rapid and severe bone loss, the fundamental reason for which is the suppression of bone formation and the enhancement of bone resorption. Studies have shown that the MG environment suppresses osteogenic differentiation mainly by decreasing osteoblast activity, and this process may be regulated by multiple ncRNAs simultaneously (Table 1). It was reported that miRNAs, lncRNAs, and circRNAs are all engaged in the regulation of osteogenic differentiation under MG exposure. These ncRNAs target different downstream factors such as ELK1 and Runx2 or affect the transcription and translation of mRNAs to influence the activity, proliferation, and apoptosis of osteogenesis-related cells, and ultimately regulate bone formation. Furthermore, the interaction between regulatory networks may enhance this effect. Therefore, targeting ncRNAs for bone metabolism therapy may be of great significance for patients who need prolonged braking and limb suspension.

## Mechanism of ncRNA regulation of osteogenic differentiation under FSS condition

FSS, as a common bone mechanical stimulus, is regarded as the principal pathway through which mechanical forces promote bone growth (Liu et al., 2022c). It was shown that the deformation of the mineralized matrix, induced by mechanical stress, leads to a non-uniform pressure gradient. This gradient, in turn, facilitates an enhanced flow of interstitial fluids within the Haversian and periosteal bone canalicular systems. Consequently, FSS is generated across the surface of osteocytes (Qin et al., 2020). Changes in mechanical loading, muscle contraction, blood pressure fluctuations, lymphatic flow, and other factors induced by physical activity could produce FSS being exerted upon bone tissue (Li et al., 2020). Appropriate FSS is crucial in regulating the function of bone formation, enhancing cell proliferation, and promoting osteogenic

differentiation (*Ding et al., 2019*; *Schneider et al., 2020*). Mechanosensitive cells are capable of perceiving mechanical signals generated by FSS within the bone environment and converting them into biochemical signals (*Sun et al., 2022b*). FSS initiates anabolic reactions in bone formation-associated cells after various signaling pathways, resulting in alterations in gene expression and an enhancement of cell differentiation (*Ding et al., 2019*).

Recent studies have shown that several miRNAs are mechanosensitive under FSS conditions. For example, upregulation of miR-132 plays a regulatory role in FSS-induced differentiation and proliferation of periodontal ligament cells (*Mao et al., 2021*). Under FSS conditions, the level of miR-33-5p was increased, facilitating the differentiation of osteoblastic MC3T3-E1 cells (*Wang et al., 2016*). In addition, FSS was able to promote F-actin cytoskeleton remodeling by inhibiting miR-23b-3p and activating the PE0 promoter leading to the upregulation of E-Tmod41 (*Peng et al., 2022*). Effector cells as well as the action parameters and duration of FSS often lead to different miRNA response mechanisms (*Peng et al., 2022*; *Wang et al., 2021b*). *Peng et al. (2022)* revealed that miR-20a expression was significantly increased in MC3T3-E1 cells 6 h post a single brief exposure to fluid shear stress (FSS) at 12 dyn/cm$^2$ for 1 h (12 h in BMSCs). Further studies showed that miR-20a may participate in the positive regulation of osteoblast differentiation by activating the BMP2 signaling pathway, achieving this through targeting and inhibiting the expression of SMAD6 and BAMBI. BAMBI is a pseudoreceptor for the BMP2 signaling pathway. SMAD6 promotes smurf1-induced degradation of RUNX2, thereby repressing activation of the BMP2/RUNX2 signaling pathway (*Mai et al., 2024*). In another study, after loading FSS at 12 dyn/cm$^2$ for 0, 30, 60, and 90 min, or at 0, 3, 6, 9, 12, 15, and 18 dyn/cm$^2$ FSS for 1 h, FSS dramatically decreased the expression of miR-214-3p and increased the expression of its target gene, ATF4, in MC3T3-E1, as well as significantly inhibited mitochondria-mediated apoptosis (*Zhang et al., 2022a*). *Huang et al. (2022)* showed that FSS induction was able to downregulate miR-140-5p to promote cell proliferation in MC3T3-E1 through activation of the vascular endothelial growth factor A (VEGFA)/ERK5 signaling pathway. VEGFA serves as a vascular growth factor, playing a principal role in angiogenesis and osteogenesis, significantly promoting bone repair and regeneration (*Hu & Olsen, 2016*). This could represent a novel pathway underlying FSS-induced osteoblast proliferation. In addition, *Wang et al. (2021b)* found that lncRNAs can be able to interact with miRNAs as ceRNAs to co-regulate the proliferation and differentiation of osteoblasts. This study demonstrated that the lncRNA TUG1/miR-34a/ FGFR1 axis is of vital importance in FSS-induced proliferation and apoptosis of MC3T3-E1 cells.

Prior research has demonstrated that stable FSS at 12 dyn/cm$^2$ for 1 h promotes osteogenic differentiation of MC3T3-E1, hPDLCs, and BMSCs (*Tang et al., 2014*). *Dole et al. (2021)* demonstrated that FSS treatment at 10 dyn/cm$^2$ for 2 h was able to stimulate OCY454 bone subdividing, a mechanism that may be associated with the activation of TGF-β, which subsequently down-regulates miR-100 and indirectly activates classical Wnt/β-catenin in osteoblasts. The Wnt signaling pathway is crucial in bone development, homeostasis, and mechanoregulation (*Warboys, 2018*). However, current studies have

**Table 2  Flow shear stress effects on differentiation and proliferation mechanisms in various cell types.**

| Evidence | FSS stimulation | Cell type | Mechanism of action | Main findings |
|---|---|---|---|---|
| Ding et al. (2019) | 12 dynes/cm$^2$ FSS for 45 mins | Osteoblast-like MC3T3-E1 | NFATc1-ERK5 signaling pathway induces E2F2 and cyclin D1 expression | FSS promotes osteoblast proliferation via the NFATc1-ERK5 signaling pathway |
| Tang et al. (2014) | 12 dynes/cm$^2$ FSS for 2 h | Human periodontal ligament cells (hPDLCs) | Activation of ERK1/2 and MAPK pathways | FSS enhances osteogenic differentiation of hPDLCs by upregulating ALP activity, OCN, and COL-I expression |
| Peng et al. (2022) | 12 dynes/cm$^2$ FSS for 1 h | MC3T3-E1 and BMSCs | BMP2 signaling via SMAD6 and BAMBI | FSS promotes osteoblast differentiation through BMP2 signaling pathway, increasing RUNX2, ALP, and SP7 expression |
| Wang et al. (2021b) | 12 dynes/cm$^2$ FSS for 0, 30, 60, or 90 mins | MC3T3-E1 | lncRNA-TUG1 regulates miR-34a and FGFR1 expression | FSS promotes osteoblast proliferation by regulating lncRNA-TUG1/miR-34a/FGFR1 axis |
| Prodanov et al. (2013) | Pulsatile fluid flow (PFF) at 0.7 Pa for 14–16 h | MC3T3-E1 | Cell membrane interaction with flow-induced nitric oxide (NO) release, affecting mRNA expression of vinculin and β-catenin | PFF promotes osteoblast proliferation and differentiation through NO release and subsequent gene expression changes |
| Elashry et al. (2021) | 12 dynes/cm$^2$ FSS for 1 h | Equine adipose-derived mesenchymal stem cells (MSCs) | FSS and mechanical stimulation of Runx2 expression | FSS promotes osteogenic differentiation of equine MSCs by enhancing Runx2 expression |
| Zhang et al. (2022a) | 12 dynes/cm$^2$ FSS for 1 h | MC3T3-E1 | miR-214-3p/ATF4 signaling pathway | FSS promotes osteoblast proliferation via miR-214-3p/ATF4 signaling pathway |
| Dole et al. (2021) | 10 dynes/cm$^2$ FSS for 1 h | OCY454 osteocyte-like cells | miR-100 inhibits Wnt/β-catenin signaling, affecting TGFβ expression | FSS regulates osteocyte differentiation via miR-100-mediated inhibition of Wnt/β-catenin signaling |

mainly focused on the osteogenic differentiation-inducing effects of FSS in MC3T3-E1 cells, with relatively few reports for other cell types. To more thoroughly elucidate the mechanism of osteogenic differentiation in response to FSS, it is vital to broaden research on the mechanism underlying FSS-induced osteogenic differentiation in various cell types. Interestingly, the application of FSS in combination with different types of biomaterials has gained more attention in recent years. It was shown that pulsed fluid on nanostructured biomaterials increased collagen and alkaline phosphatase (ALP) mRNA expression in MC3T3-E1 cells (Huang et al., 2019). FSS combined with bioscaffolding materials could promote osteogenic differentiation in equine adipose-derived MSC by upregulating ALP and Runx2 expression (Elashry et al., 2021).

In summary, FSS, as a major mechanism of mechanical force stimulation of bone growth, is capable of initiating anabolic reactions in osteoblasts via various signaling pathways, thereby promoting osteogenesis. Cumulative studies have shown that FSS can regulate osteogenesis through the regulation of various mechanosensitive ncRNAs such as miR-20a, miR-33-5p, miR-140-5p, and miR-34a. However, the regulatory mechanisms of ncRNAs in response to FSS tend to vary widely depending on factors such as the intensity of fluid pressure, the duration of application, the specific cell type involved, and the microenvironment (as shown in Table 2). Current studies have focused on cells such as

MC3T3-E1 and MSCs, and future studies could focus more on the types of effector cells and the links between the corresponding mechanisms. In addition, the application of FSS in combination with biomaterials is important for the induction effect of osteogenic differentiation.

## Mechanism of ncRNA regulation of osteogenic differentiation under mechanical stretch

Mechanical stretch is a crucial factor in bone regeneration, acting as a positive regulatory element in bone metabolism (*Wu et al., 2018*). The effects of different intensities and frequencies on osteogenic differentiation vary. For instance, 10% mechanical distraction reduces inflammation and promotes osteogenic differentiation in PDLSCs (*Sun et al., 2022a*). A 12% tensile strength closely mimics physiological loading, enhancing PDLSC proliferation and osteogenic differentiation (*Liu et al., 2017a*). The optimal osteogenic promotion at 12% tensile strength occurs at 0.7 Hz (*Wang et al., 2023*), while frequencies of 0.1 and 0.5 Hz without sustained action do not promote differentiation (*Sun et al., 2021*).

Moreover, periodic tensile stress, as a common mechanical stimulus, more realistically mimics the microenvironment in which cells *in vivo* are exposed to mechanical stimulation, which is currently the most studied mechanical force (*Xie et al., 2023*). A study was conducted to mimic the tension generated during orthodontic tooth movement (OTM) by cyclic mechanical tensile to investigate its regulatory mechanism on the induction of osteogenic differentiation in PDLSCs. PDLSCs, responsible for bone remodeling during OTM, are highly sensitive to force, which is an ideal cellular model for mechanistic studies (*Huang, Yang & Zhou, 2018*). The compressive stress generated in OTM inhibits osteogenic differentiation of PDLSCs, whereas the tensile stress generated promotes their osteogenic differentiation. *Chang et al. (2017)* utilized a cyclic mechanical stretching protocol of 12% intensity, 0.1 Hz, 5 s stretching, and 5 s relaxation. This regimen was applied for durations of 24, 48, and 72 h to mimic OTM treatment. The findings indicated that the expression of miR-195-5p was suppressed and exhibited a negative association with osteogenic differentiation of PDLCs. Further experiments revealed that miR-195-5p suppressed the activation of FGF, BMP, and WNT/β-catenin signaling pathways by suppressing the translation of FGF2, BMPR1A, and WNT3A proteins, which significantly reduced ALP activity as well as the expression of osteogenesis-related proteins, such as OSX, OPN, and OCN, and ultimately inhibited matrix mineralization. FGF2 may be involved in cyclic stretch-induced bone formation by activating the proliferation of PDLCs rather than promoting cell differentiation. Furthermore, *Meng, Wang & Wang (2022)* showed that the downregulation of miR-34a and miR-146a expression was similarly negatively associated with the osteogenic differentiation capacity of PDLCs using a cyclic mechanical tensile regimen of 10% strength, 0.1 HZ, 5 s stretching, and 5 s relaxation for 24 and 72 h. MiR-34a and miR-146 inhibit cyclic mechanical distraction-induced osteogenic differentiation of PDLSC through down-regulation of CUGBP leave-like family member 3 (CELF3). However, the underlying mechanisms of
how miR-146a and miR-34a modulate CELF3 expression remain unclear (*Meng, Wang & Wang, 2022*). Besides studies on PDLCs, researchers have also explored the mechanisms by which mechanical stretch regulates bone formation in stem cells such as adipose tissue-derived stem cells (ADSCs) and MSCs. *Li et al. (2015)* showed that mechanical stretch significantly inhibited the miR-154-5p expression and negatively regulated osteogenic differentiation of ADSCs by suppressing the Wnt/PCP pathway. This study found that miR-154-5p negatively modulates the protein level of Wnt11, inhibits RhoA activation, and reduces the expression of active ROCKII, which in turn regulates the activity of the nonclassical Wnt/PCP pathway. The findings suggest that the miR-154-5p-Wnt11-Wnt/PCP pathway might play a pivotal role as a mechanotransduction mechanism for ADSCs when subjected to mechanical stretch.

Multiple studies have found that ncRNAs can be involved in mediating mechanical stretch to promote osteogenic differentiation in a variety of ways. Previous studies revealed that miRNAs could modulate osteoblast differentiation *via* osteocytes as mediators (*Zeng et al., 2019*). Osteocytes in the bone lumen-tubule system, serve as mechanosensory cells within bone tissue. Osteocytes can transduce mechanical stimuli into biochemical signals, which in turn regulate bone remodeling by controlling the activity of osteoblasts and osteoclasts (*Thi et al., 2013*). *Zeng et al. (2019)* found that MLO-Y4 osteocytes responded to the cyclic mechanical tensile strain of 2,500 με applied at a frequency of 0.5 Hz. After exposure to mechanical stretch, the expression of miR-29b-3p in osteocytes was down-regulated, which subsequently enhanced the secretion of IGF-1. This increase in IGF-1, in turn, promoted the osteogenic differentiation of MC3T3-E1 cells. However, in MC3T3-E1, miR-29b-3p is not related to cellular mechanisms responding to mechanical tensile stimuli. In addition, mechanical stretch stimulation may achieve regulation of osteogenic genes by activating the promoters of host genes to regulate miRNA expression or by stimulating co-expression of both. Several miRNAs are situated within the intronic sections of protein-coding genes, also known as host genes, *i.e.*, "intronic miRNAs". Therefore, these miRNAs are positioned at specific genomic locations, where the majority are co-expressed with their host genes and tend to fulfill related functions. It has been shown that both miR-103a and its host gene PANK3 are decreased, while Runx2 protein expression is increased during cyclic mechanical tensile-induced osteoblast differentiation (*Zuo et al., 2015*). Thus, mechanical tensile may modulate miR-103a expression or both co-expression by activating the promoter of PANK3 to modulate Runx2 gene expression. However, since the exact role of PANK3 in osteogenesis has not been fully elaborated, it remains to be further explored whether PANK3 is also involved in osteoblast differentiation under conditions of cyclic tensile stress condition. Beyond miRNAs, lncRNAs are equally crucial in the osteogenic differentiation of osteoblasts such as PDLSCs induced by mechanical stretch (*Wang et al., 2021a*). Recent studies revealed that inducing osteogenic differentiation of PDLSCs using a 10% intensity, 0.5 Hz tensile stress regimen significantly inhibited the expression of lncRNA small nucleolar RNA host gene 8 (SNHG8), while significantly enhancing the expression of genes associated with osteogenic differentiation (*Zhang et al., 2022b*). Further studies revealed that lncRNA SNHG8 may

negatively regulate the osteogenic differentiation of PDLSCs by targeting the enhancer of zest homolog 2 (EZH2). EZH2 is a common epigenetic regulator that inhibits RUNX2 expression and osteoblast differentiation (*Dudakovic et al., 2020*).

In summary, tensile stress, as an important factor in regulating bone regeneration, is commonly used to model the tension generated during OTM. Periodic tensile stress is the most studied mechanical force. Due to the different microenvironmental stimuli that different mechanical stretch protocols can provide to the cells, the osteogenic differentiation effects and regulatory mechanisms are significantly different. It has been shown that 12% mechanical tensile strength is recognized as a good correlation with strain conditions under physiological loading conditions, which better mimics orthodontic stress and induces the proliferation and differentiation of PDLSCs. Multiple ncRNAs are engaged in mechanical tension-induced osteogenic differentiation. These ncRNAs regulate osteogenic differentiation by targeting common signaling pathways of bone formation such as FGF, BMP, and Wnt/β-catenin. Nevertheless, it has been identified that miRNAs can also regulate osteoblast differentiation not directly affecting osteoblasts but through osteocytes as a mediator. In addition, mechanical tensile stimulation may also regulate osteogenic genes by activating the promoters of host genes to regulate miRNA expression or by stimulating co-expression of both. The regulatory effect of ncRNAs on cellular osteogenic differentiation varies greatly due to the differences in tensile stimulation protocols and the mechanosensitive cells on which they act. Therefore, appropriate tensile stress protocols, including frequency, intensity of stimulation, and loading time, are important to better promote osteogenic differentiation.

## Mechanism of ncRNA regulation of osteogenic differentiation under other mechanical loading

Numerous studies indicate that vibration therapy, particularly whole-body vibration (WBV), effectively improves bone density, prevents bone loss, and reduces osteoporotic fracture risk. Vibrations are categorized into low-magnitude vibrations (LMV) with accelerations less than 1 g, and high-frequency vibrations (HFV) with frequencies over 10 Hz. Recently, LMV has gained attention as a non-pharmacologic approach for osteoporosis prevention and treatment due to its simplicity, low cost, safety, and minimal side effects.

Some studies have shown that LMV with frequencies between 20–90 Hz can effectively enhance the osteogenic differentiation of BMSCs. Some studies have also reported that vibration with acceleration lower than 1 g and frequency between 20–90 Hz can enhance the anabolism of bone tissue, thereby increasing bone mass and bone density in human and animal models (*Karinkanta et al., 2010*; *Lee, Kim & Lim, 2017*). A previous study reported that LMV with an acceleration of 0.9 g and a frequency of 45 Hz promoted osteogenic differentiation of BMSCs in de-ovulated osteoporotic rats (*Lau et al., 2011*). These studies suggest that vibration of appropriate intensity and frequency can effectively induce osteogenic differentiation behavior in BMSCs. *Yu et al. (2020)* demonstrated that LMW intervention significantly up-regulated the expression of miR-378a-3p in rat BMSCs, which in turn targeted down-regulation of growth factor

receptor-bound protein (Grb2) expression, and ultimately promoted the osteogenic differentiation of BMSCs.

In the study by Zhao et al. (2019), WBV induces bone formation by upregulating the expression of miR-335-5p, which inhibits the early action of Dickkopf-related protein 1 (DKK1). DKK1 is a typical inhibitor of the Wnt/β-catenin signaling pathway and can block Wnt signaling at an early stage (Zhao et al., 2019). Therefore, we hypothesize that variations in WBV intervention parameters, methods, and application sites may significantly affect its mechanisms of action on osteogenesis.

In summary, WBV can promote bone formation by regulating the expression of miRNAs such as miR-378a-3p and miR-335-5p and targeting related downstream factors. However, there are still fewer in-depth studies on vibration therapy to promote bone formation. More types of vibration therapy could be included for future exploration. Comparing the effects and underlying mechanisms of vibration therapy with different parameters and modes on osteoblasts could help identify optimal treatment strategies for diseases like osteoporosis.

## CONCLUSIONS

This article reviews how different ncRNAs regulate bone formation under various mechanical loadings. Bone, a highly vascularized tissue, relies on the interplay between blood vessels and bone cells to maintain integrity. Mechanical loading is crucial for bone remodeling, promoting bone formation, and preventing bone loss. Osteoblasts, key mechanoreceptors, convert mechanical stimuli into signals for bone matrix formation and mineralization. While mechanical loading regulates osteoblast proliferation, differentiation, and apoptosis, the role of miRNAs in these processes needs further exploration. Different mechanical stimuli, including tensile stress, microgravity (MG), fluid shear stress (FSS), mechanical distraction, and vibration, modulate ncRNA expression during osteogenic differentiation. This modulation affects target gene expression and bone formation. MG influences osteogenic cell activity by affecting mRNA transcription and factors like Runx2. FSS impacts osteoblast proliferation, differentiation, and apoptosis through various ncRNAs. Mechanical we hypothesize that differences in WBV intervention parameters, methods, and sites may lead to significant variations in its mechanisms of action on osteogenesis stretch regulates osteogenic differentiation by targeting pathways like Wnt and BMP via miRNAs. Vibration affects bone formation by modulating factors such as Grb2 in BMSCs at different frequencies.

However, the coordination between different mechanoreactive pathways and the key ncRNAs involved in their regulation remains unclear. It is also uncertain how varying parameters within the same mechanical response affect the ncRNAs that regulate bone formation. Given the diversity of miRNAs and their target genes, research in this area is still incomplete. Future studies should focus on identifying critical ncRNAs related to osteogenesis and examining the effects of various mechanical loadings on osteoblasts. This will help explore new strategies to regulate ncRNAs through mechanical loading, offering innovative approaches to promote bone formation and maintain bone homeostasis.

### Funding

The authors received no funding for this work.

### Competing Interests

The authors declare that they have no competing interests.

### Author Contributions

- Huili Deng conceived and designed the experiments, performed the experiments, analyzed the data, prepared figures and/or tables, authored or reviewed drafts of the article, and approved the final draft.
- Dongfeng Wan conceived and designed the experiments, authored or reviewed drafts of the article, and approved the final draft.

### Data Availability

This is a literature review.

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
