# Peer review of "Mechanical loading regulates osteogenic differentiation and bone formation by modulating non-coding RNAs"

_PeerJ, doi:10.7717/peerj.19310_

## Round 0.1 · original submission · Major Revisions

The reviewers acknowledge the merits of this study. However, they also pointed out some issues to be improved with your manuscript. Please revise it according to the reviewers' comments. We hope that the reviewers' comments will help you improve your paper.

Reviewer 1 ·

Basic reporting

This review article discusses how mechanical loading influences bone formation through the modulation of non-coding RNAs (ncRNAs). The topic is cutting-edge and highly relevant. The article is well-structured and provides a comprehensive review of the role of mechanical loading and ncRNAs in bone formation.However, there are certain aspects that could be strengthened, particularly regarding the breadth of literature citations, the clarity of some sections, and the overall logical flow. Below are the detailed reviewer comments:
Language Expression and Specific Detail Issues:
(1) In line 22, “Components of bone tissue is a hard and dense connective tissue” should be corrected to “are,” as bone tissue consists of multiple components, and using the plural is more accurate.
(2) In line 26, “Bone can respond to stimuli from mechanical loading” followed immediately by “Physiological loading can induce bone formation” is semantically repetitive. It is recommended to combine these two sentences for conciseness.
(3) In line 114, please capitalize the first letter of "circRNA."
(4) While the article is written in a professional tone, some sentences are overly long and not concise. It is suggested to optimize the sentence structure for clarity and brevity. Additionally, some references need formatting adjustments to ensure consistent citation style throughout the manuscript.

Experimental design

(1) The overall structure of the article is relatively clear, but some sections could be further condensed and better organized, especially the portions discussing the functions of miRNA, circRNA, and lncRNA. It is recommended to merge some content to avoid redundancy and streamline the flow.

(2) The descriptions of the research methods used in the cited studies are somewhat brief, particularly in the construction of ceRNA networks. There is a lack of explanation regarding experimental design, sample selection, and experimental validation. It is recommended that the authors include more detailed experimental methods or refer to relevant methodologies in the literature to increase the credibility and transparency of the review.

(3) While many studies are cited, there is a lack of in-depth discussion of specific experimental results. In particular, how ceRNA networks are validated through experimental data is not adequately addressed. It is suggested to include more data and experimental evidence to strengthen the scientific rigor of the article.

(4) The article mentions that miRNAs compete to bind target genes, but does not elaborate on the mechanism by which target genes are selected. It would be beneficial to include more detailed content on how miRNAs choose their target genes, particularly whether there is specificity in target gene selection during bone formation.

(5) The review discusses the role of circRNAs in bone remodeling but does not provide a detailed explanation of how circRNAs regulate the miRNA-mRNA axis during osteogenesis. A more in-depth exploration of the specific mechanisms by which circRNAs interact with miRNAs would greatly enhance this section.

(6) The article categorizes ncRNA changes caused by different types of mechanical forces. It would be helpful to provide a brief comparison of these mechanical forces and discuss the differential regulatory effects of ncRNAs under various loading conditions.

(7) The title refers to the regulation of "bone remodeling by ncRNAs," but bone remodeling is the result of both osteoblastic and osteoclastic activities. Most of the article, however, focuses on osteogenic responses. It is suggested to either narrow the focus of the review or include a discussion of how ncRNAs also participate in osteoclastic activity.

Validity of the findings

While the conclusion summarizes the role of ncRNAs in mechanical loading, it lacks a concrete outlook on future research directions. It is recommended that the conclusion also discuss how these findings can be applied to advancing treatments for bone diseases or promoting bone regeneration in clinical settings.

Reviewer 2 ·

Basic reporting

1. The Title of the manuscript need to be corrected. For example , "Mechanical loading regulates osteogenic diûerentiation and bone formation by modulating non-coding RNAs" maybe better.
2. The abstract of the manuscript need to be improved. For example, in the last sentence,”enhance
osteogenic diûerentiation” should be corrected into "regulate osteogenic diûerentiation or bone formation".
3. The overall structure of the manuscript needs improvement,because the structure of the manuscript
is quite scattered. It is better that the manuscript contains “microRNA”,“LncRNA”,“Circ
RNA” parts.
4. Many sentences and sections are unnecessary and need to be deleted and revised. For example ,page 12-13, from line 199 to line 230, page 18, from line 353 to line 363, page 21, from line 438 to line 443, page 22, from line 467 to line 472 .

Experimental design

1.In the manuscript,some " mechanical tensile stress " should be cerrected to " mechanical stretch". The author should check the references carefully.
2.Page 22,"we hypothesize that differences in WBV intervention parameters, methods, and sites may lead to significant variations in its mechanisms of action on osteogenesis" was coufused,which should be corrected or deleted."Comparing the effects and intrinsic mechanisms of vibration therapy interventions with different parameters and modes of action on osteoblasts will provide ideas for finding optimal treatment options for diseases such as osteoporosis." also was confused.
3. Page 10-12, the manuscript shoud provide more detailed imformation of LncRNA-miRNA-mRNA regulatory network.
4. Please provide more imformation of "paraspeckles".
5. Figure 1 should be deleted, and Figure2 should be improved.

Validity of the findings

no comment

Additional comments

no comment.

---

## Round 0.2 · accepted · Accept

I have confirmed that the authors have addressed all reviewer comments, and therefore I accept this paper for publication in PeerJ.